# Mode I Fatigue of Fibre Reinforced Polymeric Composites: A Review

**DOI:** 10.3390/polym14214558

**Published:** 2022-10-27

**Authors:** Xingzhong Gao, Muhammad Umair, Yasir Nawab, Zeeshan Latif, Sheraz Ahmad, Amna Siddique, Hongyue Yang

**Affiliations:** 1School of Textile Science and Engineering, Xi’an Polytechnic University, Xi’an 710048, China; 2Key Laboratory of Functional Textile Material and Product, Xi’an Polytechnic University, Ministry of Education, Xi’an 710048, China; 3Ministry of Education Key Laboratory for Advanced Textile Composite Materials, Tiangong University, Tianjin 300387, China; 4Department of Textile Engineering, School of Engineering and Technology, National Textile University, Faisalabad 38000, Pakistan; 5Department of Textile Technology, School of Engineering and Technology, National Textile University, Faisalabad 38000, Pakistan

**Keywords:** mode I, fatigue behaviour, fibre reinforced polymer composite, strength improvement

## Abstract

Composites are macroscopic combinations of chemically dissimilar materials preferred for new high-tech applications where mechanical performance is an area of interest. Mechanical apprehensions chiefly include tensile, creep, and fatigue loadings; each loading comprises different modes. Fatigue is cyclic loading correlated with stress amplitude and the number of cycles while defining the performance of a material. Composite materials are subject to various modes of fatigue loading during service life. Such loadings cause micro invisible to severe visible damage affecting the material’s performance. Mode I fatigue crack propagates via opening lamina governing a visible tear. Recently, there has been an increasing concern about finding new ways to reduce delamination failure, a life-reducing aspect of composites. This review focuses on mode I fatigue behaviours of various preforms and factors determining failures considering different reinforcements with respect to fibres and matrix failures. Numerical modelling methods for life prediction of composites while subjected to fatigue loading are reviewed. Testing techniques used to verify the fatigue performance of composite under mode I load are also given. Approaches for composites’ life enhancement against mode I fatigue loading have also been summarized, which could aid in developing a well-rounded understanding of mode I fatigue behaviours of composites and thus help engineers to design composites with higher interlaminar strength.

## 1. Introduction

Fibre reinforced composites are now used for emerging structural application areas such as aerospace, automotive, wind turbine, defense area, marines, civil engineering and leisure equipment as replacements for conventional metallic materials [1,2,3,4,5]. Glass and carbon fibres are the two most common reinforcing components which are used in fibre reinforced plastic (FRP) composites. The combination of glass and carbon reinforcements with polymer-based matrix results in a wide range of materials that exhibit lighter weight, high strength, hardness, tolerability, toughness, stiffness and are more resistant to corrosion and environmental factors. These FRP composites are heterogeneous and display anisotropic behaviour in nature. As the FRP composites are made up of fibres and matrix, their mechanical properties are quite different from metallic materials because the mechanical performance of FRP depends on their constituents. The mechanical properties of FRPs are primarily based on the strength, modulus and chemical stability of the fibre and matrix [6].

Compared to other damage types, delamination growth is a prevalent life-reducing failure mode for composite structures causing the separation of adjacent plies [1,4,7,8]. The interlaminar cracks initiate and grow between neighbouring layers under static or dynamic conditions [9,10]. This happens due to low toughness and cyclic strength of resin-rich areas between the adjacent layers of laminate. Delamination initiates and propagates under the recurring use of quite low interlaminar fatigue loads [11,12]. The lower values of interlaminar fatigue loads lead to complete collapse of composite structures, and consequently it can be a life-threatening structural reliability problem. In carbon fibre reinforced epoxy composites, this issue becomes more thought-provoking since the cyclic cracks’ propagation cannot be sensed using visual inspection. With increasing demand for composites in critical structural components, it is essential to ensure damage tolerance to increase safety.

A damage tolerance design approach assumes that defect already exists in the structure and permits valuation of the time to failure. Interlaminar fatigue characterization of composites is essential to providing valuable input for damage-tolerant design philosophy. It is employed in the testing for the cyclic performance of materials since three dimensional (3D)-woven composites are used in lightweight structural components, beams, structural joints, and wind turbine blades. Apart from this, one of the most critical applications of 3D composites is in the Boeing Dreamliner airliner (in the inlet fan blades of the GenX gas turbine engines). In all these applications, composites are exposed to cyclic or variable loading conditions, so knowledge about fatigue response, including delamination resistance to the initiation and growth of cracks, is typically important.

### Delamination Modes

Delamination propagates under the three pure modes: mode I (opening mode), mode II (sliding mode) and mode III (tearing mode) or as a combination of these [4]. Fatigue delamination is a general kind of failure of structures in composites [13], so it must never be ignored in any design process. Pure mode I and II tests are mostly used to characterize composites in fatigue [14,15,16]. Under mode I loading, the delamination evolution typically takes place in a direction normal to the loading [1]. Stress energy release rates (SERR) and intensity factors are used to quantify the delamination properties of composites under fatigue loading. In the literature, the proposed delamination threshold and propagation models under mode I fatigue loading are based on the Paris power law equation, which was formerly stated by means of the difference in maximum and minimum values of mode I stress intensity factors [17]. It describes the relationship between SERR and fatigue crack growth as
(1)$$\mathrm{da}/\mathrm{dN}=A{\left(\Delta K\right)}^{m},$$
where
a = crack length (delamination length);N = number of cycles;da/dN = fatigue crack growth rate;ΔK = difference in the Mode I stress intensity factors (maximum and minimum);A, m =material constants for curve-fitted power law.

According to the Paris law given in Equation (1), the cyclic crack evolution rate depends only on the stress intensity factor range (ΔK). Many researchers have found that the strain energy release rate (G) is a more appropriate parameter to characterize the mode I delamination in composites instead of the stress intensity factor that is used for metals [18].

In composites, the fatigue damage and failure processes are more complex than in metallic materials where a crack develops and proceeds until fracture. Many microcracks start at the early phase of the fatigue formation in the composite and as a result, different kinds of fatigue damage progress. In case of composite laminates, fatigue delamination can lead to catastrophic failure [19]. Fibre volume fraction is a dominant factor in explaining the composite laminates’ fatigue behaviour. The fatigue resistance improves by increasing the fibre volume fraction at a specific point, and then reduces because of the shortage of sufficient resin to hold the fibres. Apart from this, there are different parameters to be considered for composite fabrication to ensure required damage tolerance, such as material components, fibre direction, hysteresis heating, kind of load, interface characteristics, mean stress and environmental conditions.

In composite materials, crack nucleation and growth arise mostly due to stress concentrations encouraged by imperfections such as holes, microcracks and inhomogeneities that occur in the fabrication process or during use. The fatigue performance is largely influenced by flaws as they depreciate the quality of the adhesive interface. Hence, the fatigue performance somewhat depends of the excellence of the composite fabrication process [20].

The various fatigue loading modes of composites include Tension–Tension (T-T), Compression–Compression (C-C), Tension–Compression (T-C), bending, thermal and mixed mode loadings [21,22]. Various loading conditions cause various damage processes. Many studies are concerned with the fatigue response of the composites. In some of the studies, the fatigue damage of structural composite has been investigated in the context of various loading conditions [23,24,25,26,27].

It is assumed that crack starting and crack proceeding are the two stages that occur during fatigue. Extreme crack starting points are obtained first in cyclic loading, and a large part of the fatigue life is crack proceeding at high fatigue stress generally known as low cycle fatigue (LCF). In high cycle fatigue (HCF), the fatigue crack must be initiated after a significant number of total fatigue cycles. In metal, fatigue initiation is obtained at the surface where defects or stress concentration are most likely to be found. In composites, fibre/matrix heterogeneity of matrix gaps can produce the defect point and thus it is not unusual to find the subsurface fatigue origins.

Fibre fracture, matrix fracture and fibre–matrix delamination are the three dominant fatigue failure modes observed in unidirectional single layer composite. The fibre strength has a considerable effect on fibre fracture. The high stiffness of the fibres can delay the matrix fracture because a lesser amount of deformation passes to the matrix. Delamination occurs when the fibre–matrix interface is simulated by weak fibre/matrix bond strength. In brief, various damage mechanisms interact, hence it is quite challenging to analyse the fatigue failure in composites. This is true for unidirectional as well as multidirectional textile composites.

Considering these challenges, in this paper we have provided a review of the investigations that have been made on mode I fatigue loading of fibre reinforced composite. Sufficient information on fatigue behaviour and failure modes of various preform structures is given, which could help to adopt a defect tolerant approach both in design stage and in residual life estimation. Several parameters that affect the performance of composites under mode I fatigue loading such as temperatures, load rate and effect of fibre bridging are described briefly. Methods to improve the fatigue performance of composites are also summarized.

## 2. Mode I Fatigue Behaviour of Various Preforms Structures Used in Textile Composites

### 2.1. Unidirectional Composites

Most of the FRP composites are made from laminates. Among various composite structures, composite laminates’ fracture is the most common. Figure 1 is the schematic presentation of unidirectional laminated composites. Due to absence of reinforcement in thickness way, they suffer from delamination occurrence. This weak point can cause delamination spread between neighbouring layers under either fatigue or quasi-static loading. It has been broadly declared that delamination causes critical damage in laminated composites and is necessary to consider in composite structural design [1,2,4,8,20,28,29,30,31,32,33,34]. For example, in a study on carbon–epoxy laminated composites, Brunner et al. [11] determined the fatigue threshold SERR for initiation of delamination under mode I loading to be as low as ~60 J/m^2^. Y. Nakai and C. Hiwa [35] examined the fatigue crack formation under mode I loading with two types of unidirectional carbon fibre/epoxy resin laminates (M4J/2500 and T300/3601). They highlighted the effects of environment and loading frequency on fatigue delamination crack growth. In the first type of M4J/2500 laminate, mode I fatigue delamination rate was found to be cycle-dependent for both air and water and the fatigue crack growth rate in the water (Figure 2) was lower than that in the air (Figure 3). In the second type T300/3601 laminate, mode I fatigue delamination rate in the air was dependent on the number of cycles while in the water it was dependent on time, as shown in Figure 4 and Figure 5, respectively [35].

Kenane, M. et al. [36,37], conducted fatigue characterization of unidirectional glass-fibre/epoxy-resin laminates under mode I loading conditions for different combinations of energy release rate mode ratio. The constants in the Paris equation were found for each $G_{\mathrm{II}}/G_{\mathrm{th}}$ taken as a model ratio. A double cantilever beam (DCB) test was used to analyse the mode I fatigue delamination of the composite samples. A scanning electron microscope (SEM) was used to unveil the fatigue delamination fracture modes as shown in Figure 6 [36]. They reported the fatigue threshold values of strain energy release rate ($G_{\mathrm{th}}$) under which delamination development is not initiated. $G_{\mathrm{th}}$ was found to be increased with increase in the $G_{\mathrm{II}}/G_{\mathrm{th}}$ ratio. SEM images of fractured surfaces were macroscopically smooth with huge quantity of fibre breaks and resin fragments.

Vauthier et al. [38] studied the influence of ageing in hygrothermal conditions on the fatigue behavior of the unidirectional glass-fibre/epoxy-resin composite. In the first step, different factors including moisture and temperature were used to analyse the fatigue behaviour of unaged composite. At elevated temperatures, the relation between the surrounding moisture and crack tip was seen to induce significant losses in lifetime. In the second step, fatigue behaviour was examined after the initial ageing step. The presence of hygrothermal defects in the bulk composite during the water sorption process resulted in a significant reduction in fatigue characteristics, particularly after ageing by water immersion. The overall losses of the fatigue characteristics were connected to a decrease in the statistical distribution of the fibre strength after ageing. The micro-mechanical test of the first fibre break was analysed during mechanical loading [38]. Fatigue behaviour in unidirectional composites under tension–tension fatigue loading was generally found as a function of the fibre characteristics of composites and the arrangement of the fibre from the loading direction. At the initiation of loading of the composite with fibres arranged in the direction of loading, the cracks in the matrix appeared along the fibre direction [39].

J. Hoffmann and G. Scharr analysed the fatigue resistance of composites having two different geometric shape pins (rectangular and circular shape). According to the Paris curves, the rectangular z-pins were more efficient in growing delamination resistance of quasi-isotropic composites unlike the circular z-pins. In unidirectional composites, when the cross section of the pins changed, it slightly affected the mode 1 fatigue resistance of the z-pins composite.

### 2.2. Multidirectional Laminated Composites

Multidirectional laminated composites are those in which the layers are arranged at different angles to produce multidirectional stiffness properties for broad area uses such as quasi-isotropic laminates as shown in Figure 7. These are made up of a combination of layers, e.g., at 0°, 45°, −45° and 90° angles that have isotropic in-plane elastic moduli. Delamination growth in multidirectional laminates generally consists of several delamination cracks which frequently develop and transfer into distinct ply interfaces by cracking of resin. Therefore, delamination does not constantly persist at the primary plane for laminated multidirectional composites [40]. Delamination migration is a common occurrence in multidirectional laminated composites and can be used in a variety of loading scripts and parts [41]. Delamination relocation (migration) is the consequence of merging of a succession of angled microcracks that grow from the crack tip on ward.

These microcracks combine and propagate in out-of-plane direction until they reach a favourable interlaminar interface. The development of composite structure requires extensive knowledge of the movement process, particularly similarities and differences between fatigue and quasi-static delamination movement.

Peng et al. [42] studied the certain property of ply orientation in multidirectional carbon with bismaleimide composites under Mode I fatigue loading. A double cantilever beam (DCB) test was used to analyse the effect of ply orientation on fatigue cracking rate and threshold value rate under mode I fatigue loading. It was observed that the fatigue growth rate was based on stabilized interlaminar energy $G_{I}/G_{\mathrm{IC}}$ instead of $G_{I}$ (mode I strain energy release rate) by growing the delamination resistance effect of the sample. It was reported that the strain threshold was nearly equal to $G_{\mathrm{th}}$ (threshold value for fatigue crack growth); to the corresponding $G_{\mathrm{IC}}$ (mode I critical) was $G_{\mathrm{TH}}/G_{\mathrm{IC}}\;,$ which was constant and unaffected by fibre direction in the mid-plane. Moreover, it was also observed that fibre-bridging and internal cracking were the main factors in increasing delamination fatigue resistance [42].

T Chocron et al. studied the fracture and fatigue delamination growth in multidirectional laminated composites under mode I fatigue loading. A Paris-type relation was obtained by fatigue tests of DCB specimens, which describes the delamination growth rate where a is the delamination length and N is the cycle number; displacement ratio of R = 0.10 and 0.48 were used [43]. Banks-Sills et al. [44] studied the fatigue delamination behaviour of carbon fibre/epoxy resin composites manufactured by different methods but with the same stacking sequence. One was made up of woven prepreg layers that were alternated with tows in the direction of 0°/90° and +45°/−45° as shown in Figure 8a. The second was made up by using a wet-layup technique with a similar multi-direction as the first method. Fatigue delamination resistance tests were performed, as shown in Figure 8b. It was observed that the interface energy release rate of wet-layup was significantly lower at the initial value, but the steady-state values were similar. Fatigue delamination experiments were carried out at different cyclic R-ratios. The experimental values of da/dN were used to measure the delamination rate. R-ratio affects the da/dN curves as predicted. They reported that the fatigue propagation rate in prepreg was higher than the wet layup [44]. Delamination migration is crucial and is a special attribute of multidirectional laminated composites [41].

Gong et al. [41] studied the delamination behaviour of specifically designed composites with +θ/−θ stacking sequence under mode I fatigue loading. DCB test setup was used for fatigue characterization. Delamination through intralaminar ply separation was detected and that was verified by X-ray tomography reports. Delamination develops via +θ and −θ ply sheets sequentially until it meets the 0° ply, which stops further delamination growth. Fractographic findings from the SEM revealed the smooth fracture surfaces.

### 2.3. 3D Woven Composites

Due to the greater mechanical properties, 3D woven composites have gained a lot of attention in many application areas such as aerospace, civil engineering, marine, transportation and automotive industries [2,4,26,45,46,47,48]. In 3D woven fabrics, the through-thickness reinforcement known as z-binder yarns have a significant positive influence on mechanical performance of composite [1,49]. Mostly glass and carbon continuous yarns are used as z-binder yarns. Stegschuster et al. [50] examined the mode I fatigue delamination of thin 3D woven composites which has z binder yarns in through-thickness direction (Figure 9a). They reported that the fatigue delamination resistance increased significantly as the volume fraction of z-binding increased as shown in Figure 9b. In thin non-crimp 3D woven composite, the z-binding yarns were angled at a sharp angle around 70° from the orthogonal. The sharp angle had a significant impact on crack linking stress, deformation energy absorption and breaking process in mode I pull-off experiments on a single woven z-binding yarn as shown in Figure 10.

The minimum value of cyclic stress intensity factor (${\Delta G}_{I}$) to start crack propagation was also found to increase linearly with increase in z-binder yarn volume content. It was reported that at maximum volume content of z-binder yarn (14%) the fatigue threshold value increased by 8 fold that was (>400 J/m^2^) [50]. This reported value of fatigue threshold is much greater than unidirectional, multidirectional (Table 1), but similar or greater than reported values of fatigue threshold obtained by techniques such as z-pinning, stitching or matrix toughening as given in the next section (Table 2). The enhanced cyclic delamination resistance of 3D woven composites has increased their potential being used in critical fatigue structural components. As mentioned earlier, carbon and glass yarns are commonly used as z-binders in 3D woven composites. These binder yarns increase the fatigue and fracture resistance of 3D reinforcement by formation of fibre bridging in the crack growth path. Considering this fact, Abbasi et al. [51] used metallic z-filaments to further increase the mode I fatigue properties.

### 2.4. Hybrid Composites

Ladani et al. [57] developed a 3D novel self-healing hybrid composite presenting a distinct blend of high resistance to delamination and in-situ healing of fatigue-induced cracks under mode I and mode II loading. The composite comprising two types of z-binder yarns, i.e., self-healing poly [ethylene-co-methacrylic acid] (EMAA) and high-performance carbon showed a significant rise in fatigue threshold strain energy release rate range to start cracking as well as repairability to cracks induced by fatigue loading. Recently, many studies on carbon–epoxy composites reported that EMAA is very effective in repairing delamination-induced damage [58,59,60]. Moreover, the 3D hybrid composites offer higher cyclic resistance to cracking under mode I loading as compared to mode II loading.

## 3. Failure Modes in Composite under Fatigue Loading

Fibre failure and matrix failure are the two major damage processes often found in composite during cyclic loading [36,61,62]. The analysis of fatigue damage processes provides the weakest micro-structural component, which is helpful to enhance the service qualities. The damage formation in cross-ply fibre reinforced composites exposed to cyclic tensile–tensile loading is shown in Figure 11. Figure 12 represents the failure modes observed in the GFRP composite during tension–tension fatigue loading [62]. SEM images of multidirectional carbon/epoxy composite samples after mode I fatigue loading are given in Figure 13, obtained at different crack lengths. The fractured surfaces show the presence of hackles and fibre prints. Fibre prints are the evidence of fabric–matrix debonding, and hackle formation indicates occurrence of shear stress state due to fibre pull-out [63,64].

### 3.1. Failure in Fibres

Failure of fibres in composite static or fatigue failure can be categorised as tensile and compression fibre failure [62]. Fibre pull-out, fibre fracture and debonding are the processes of fibre failure observed in mode I fatigue loading of composites [31]. Figure 14 is the schematic presentation of microcrack initiation in matrix under mode I fatigue loading. According to Khan et al., formation of fibre imprints is the indication of crack tip or delamination growth by application of load cycles [31]. It is assumed that in case of crack tip growth, fibre failure occurs behind the crack tip. When both fibre and matrix are brittle, then fibre pull-out failure is also observed [65]. Local fibre fracture has been observed in the initial loading followed by stress readjustment for the composite in tension. The fibre debonds from the matrix, which is followed by the breakage leading to the final failure [62]. Compression failure of fibres found to be less dependent on the strength of the fibre and more dependent on the fidelity of fibres such as kinking and fibre microbuckling. The area in the void surroundings and free edges is the point where fibre microbuckling generally starts. Disorders in fibres were influenced by the compressive failure of fibres. It was found that in unidirectional composite, 0.25° disorders of fibres can decrease the compressive strength by as much as 70% of its initial value [62].

### 3.2. Failure in Matrix and Interface

Failure in the matrix during fatigue loading can be divided into two failure modes, the first one is matrix failure in a ply known as inter-fibre fracture and the second one is matrix failure in between piles known as delamination. There has remained always a great interest in developing the better fibre–matrix interfacial properties [48]. Inter-fibre fracture generally initiates at the fibre–resin interface then proceeds to the resin. On the contrary, the delamination is produced by interlaminar stresses because of microcracks in the matrix. Free edges of the multidirectional composites usually generate interlaminar shear stress individualities that start microcrack [62]. Figure 14a is the schematic presentation of fibre disbonding and initiation of microcrack in matrix. Due to heterogenous nature of composite materials, interfacial strength of fibre–matrix (adhesive strength) is assumed to be lower than the cohesive strength of matrix. Therefore, by mode I fatigue loading of composite, the fibre disbonds from matrix earlier than matrix breakage. After a certain growth of disbond, it produces stress concentration in matrix. The resulting three-dimensional stress state generates microcrack in the matrix [31].

A microcrack begins to develop when the applied load is strong enough to induce fibre failure [21]. These microcracks finally result in hackle formation. In case of fibre–matrix decohesion, in addition to hackle formations on the fractured surface of samples under mode I fatigue loading, striation formations have also been observed in the fibre imprints (Figure 14b) [31]. According to Franz [66], the striations are formed in matrix at fibre–matrix interface due to extension of microcracks. However, Greenhalgh, M. [67] suggested that striations are formed as a consequence of molecular chains breakages.

Fibre failure, matrix failure, delamination and fibre–matrix separation are among the primary failure modes of laminates. The failure condition of the polymer matrix composites could be brittle [68] or ductile based on the chemistry and curing [21] agent utilised. The ductile mode is usually found by shear yielding. It is observed that the rise in brittleness of the polymer is a transition from shear yielding to micro-gaps followed by crazing [21]. Shear yielding is a type of energy absorption process connected with polymer failure. It takes place when the localised plastic flow begins as a result of the applied stress. That plastic flow may distribute the shear bands across the specimen, absorbing a considerable amount of energy or resulting in locally produced shear bands.

## 4. Factors Affecting Mode I Fatigue Behaviour

### 4.1. Effect of Fibre Bridging

In case of fibre reinforced polymeric composites, fibre bridging performs a vital role in fatigue delamination resistance. After a certain growth of delamination length, the crack development decreases significantly due to crack shielding effect of bridging fibres [69]. The bridging fibres restrict the crack growth and intermittently store and release a significant amount of strain energy in fatigue loading. Hence, the fibre bridging impedes the crack development under fatigue loading. In case of longer crack length, the fibre bridging effect becomes more noticeable. L. Yao demonstrated the effect of amount of bridging fibres on the Paris relationship, which is used to quantify the effect of fibres bridging in mode I fatigue loading.

A single Paris resistance curve has been used to evaluate the fatigue crack expansion, which is found to be ineffective in case of bridging fibres, as the bridging fibres can vastly increase delamination resistance [69]. A theoretical Paris-type relationship was introduced to clarify the role of bridging fibres in mode I fatigue delamination development by correlating amount of fibre bridging with curve fitting parameters C and N, whereas C (Paris constant) and n (amount of fibre bridging) were taken as the constants in the previous research. The general form of the Paris relation accounting for bridging fibres can be summarized as Equation (2).
(2)$$\mathrm{da}/\mathrm{dN}=C\left(a-a\_0,R\right){\Delta G}^{\left(N\left(R\right)\right)},$$
where
$a-a^{0}$ represents crack extension length;da/dN is crack development rate;ΔG is strain energy release rate range;R is stress ratio and C is the compliance of the DCB specimen. 


Here, N is independent on the fibre bridging volume but reliant on the stress ratio. However, the C is dependent on both the volume of bridging and the stress ratio. In another study, Yao et al. emphasised the importance of fibre bridging, studying the effect of interface configuration on the fatigue delamination of composites. Their research findings indicate that fibre bridging becomes more obvious in multidirectional interface.

Yao et al. [70] also analysed the influence of stress ratio on fibre bridging in the formation of composite materials under the mode I fatigue delamination. They examined and contrasted the various stress ratios of the fatigue resistance curve (R-curve) with the quasi-static R-curve. They found that the high stress ratio of the fatigue resistance curve was equivalent to the quasi-static performance. Moreover, the low stress ratio of fatigue resistance was less than the quasi-static resistance. Their results depict that fibre bridging was dependent on the stress ratio. In case of high stress ratio delamination, more bridging of fibres can be developed as compared to the low stress ratio. They concluded that fatigue bridging law was dependent on stress ratio and fatigue delamination was sequentially dependent on the load block.

To describe the impact of significant fibre bridging on fatigue delamination of multidirectional laminates, both the Paris relation and the modified Paris relation (with a modern similar variable) has been used. Bridging fibre seems to be typically an artefact of unidirectional laminated composites’ testing. It happens in laboratory DCB tests, but hardly detected and reported for delamination in actual working conditions. It makes sense that to design a structure for functioning life, the required cyclic delamination data should be conservative in that it eliminates fibre bridging [71]. The authors reported that the modified Paris relation can be accounted as a universal form of the Paris relation, which is equally valid for delamination with and without fibre bridging. It was found that during fatigue delamination of multidirectional laminated composites, the energy release remained constant with the growth of fibre bridging. In other words, bridging fibres only had influence on the energy release at the start of fatigue cycles. In many cases, these bridging fibres just stored and released strain energy intermittently but had no actual effect on permanent energy release [63].

### 4.2. Effect of Temperature

Coronado et al. [72] investigated the effect of temperature on mode I fatigue delamination of two aeronautical grade composite material at various temperature ranges of 90, 20 and −60 °C. Two individual epoxy matrices and the same unidirectional carbon fibre were used. In another study [53], the same group of authors reported the effect of temperature on failure morphologies of composite under mode I fatigue loading. These studies reported that fatigue delamination behaviour of composites depends on temperature as well as type of matrix used. It was concluded that at low temperature the composite samples under investigation became more brittle, and as the temperature raised, the matrix ductility increased. It was found that the effect of temperature with non-modified epoxy resin was more prominent. Shindo et al. [68] also described the effect of temperature on the fatigue delamination growth rate of composites. A double cantilever beam (DCB) was used to analyse the mode I fatigue delamination at room temperature, liquid helium temperature and liquid nitrogen temperature 4 K and 77 K, respectively. At low temperature, the fatigue delamination growth (mode I) was much lower than that at room temperature. At room temperature and low temperature, the main mode I fatigue delamination growth processes were distinct. The fracturing process at room temperature was fibre–matrix debonding. Both fibre–matrix debonding and brittle failure of the matrix were the most common causes of mode I fatigue delamination formation.

A. Sjogren and L.E. Asp [73] examined the effect of temperature on mode I fatigue delamination rate in the composite. For both room and high temperature, experiments were performed under mode I static and fatigue loading. For the static experiments, the critical energy release rate was slightly higher than the strain energy release rate for the delamination growth under fatigue loading. In the static experiments, at high temperatures, the threshold parameters in the fatigue loading were just around 10% of the critical parameters. The fracture surface produced at high temperature was approximately equal to the fracture surface produced at room temperature according to the fractographic study of delamination growth.

### 4.3. Effect of Load/Stress Ratio

Androuin et al. [74] investigated the impact of load ratio on fatigue crack formation at a constant and variable amplitude loading. The purpose of the research was to produce a slow-growth fatigue delamination technique for a composite that would provide moderate and reliable performance. The crack driving force (CDF) was used to analyse the overlap delamination formation curves from various loading ratios for mode I fatigue delamination. The impact of loading ratio on crack formation curves enables the development of a crack formation master curve for mode I fatigue delamination of composites.

Andersons et al. [75] introduced an analytical model for the impact of stress ratio on fatigue delamination development rate in composites through mode I loading. The model was supported by empirical loss estimation factors before the fracture. Loss calculation was based on the linear average accumulative principle. It provides for the calculation of fatigue delamination formation rate within random stress ratio using experimental reports at R-value to determine the model parameters. Within mode I load in unidirectional graphite, glass and alumina fibre and polymer matrix composite the precision of forecast was seen to be very high.

Yao et al. [70] analysed the influence of stress ratio on fibre bridging in the formation of composite materials under the model I fatigue delamination. The high stress ratio of the fatigue resistance curve was equivalent to the quasi-static performance. Moreover, the low stress ratio of fatigue resistance was less than the quasi-static resistance. Fibre bridging was found to be dependent on the stress ratio. In the fatigue delamination at the high stress ratio, more bridging of fibres occurs as compared to the low stress ratio. They concluded that fatigue bridging law was dependent on stress ratio and fatigue delamination was sequence dependent on the block load.

Argüelles et al. [20] studied the effect of loading rate on the mode I fatigue delamination initiation and propagation in unidirectional carbon–epoxy composite at various loading rates (R = 0.2 and 0.5). At higher stress level (R = 0.2), fibre–matrix damage increased both at crack initiation and propagation stage and crack path was mainly oriented on the fibre–matrix interface. However, at low stress level, less fibre debonding was reported.

## 5. Numerical Studies on Mode I Fatigue Delamination

Apart from experimental work, there are some numerical studies on Mode I fatigue delamination of composite. Fracture models are mandatory to forecast fatigue life and create appropriate scrutiny intervals to found and fix the delamination before it becomes life-threatening or exceeds the residual strength of the component [76]. Various numerical methods have been developed to achieve this purpose.

To analyse the crack propagation process and to extract the strain energy release rate at crack tip, a technique known as “Virtual Crack Closure” (VCCT) technique has been used. The method is based on the supposition that the energy ΔE released when the crack is extended by Δa from a (as shown in Figure 15) is equivalent to the energy required to close this crack [77]. The projected path of spread must be identified in advance. The crack path is physically prolonged or closed to complete finite element analysis as shown in Figure 16. For a crack modelled with 2-D elements, the work ΔE required for crack closure along one element side can be found [78] as
(3)$$\Delta E=\frac{1}{2}\left(X_{1l}\Delta u_{2l}+Z_{1l}\Delta w_{2l}\right),$$
where $Z_{1l}$ and $X_{1l}$ are the opening and shear forces at nodal point l to be closed and $\Delta u_{2l}$ and $\Delta w_{2l}$ are the differences in opening and shear nodal displacement at nodal point l as shown in Figure 15.

The energy release rate can be obtained as
(4)G=ΔE/ΔA,
where ΔA is the crack surface created. The strain energy release rate can be calculated using plate/shell elements or 3D solids. Problems of arbitrarily shaped delamination front can be solved by defining a local crack tip coordinate system. Adrian assessed the capability of commercial FE code with implementation of VCCT. The results obtained by VCCT technique demonstrate excellent agreement with previous results. The strain energy rate increased for delamination propagation up to 19 mm, and it decreased for delamination propagation higher than 19 mm. This difference is due to the change in crack growth path from unstable to stable. The distribution of strain energy released rate across the delamination front changed slightly with delamination length. The distribution became more curved at delamination lengths near the load pin.

To apply VCC technique, the expected crack growth path must be predicted. This method is commonly used in 2D finite element models with one or two crack fronts. The use of extended finite element model (XFEM) permits simulating intralaminar and interlaminar crack development. Contrary to the VCCT method, XFEM does not need a pre-defined crack path. This technique is effective in simulations where crack growth is not already known [79]. Enriched parts of displacement field were introduced to model the weak or strong discontinuities in XFEM method [80]. Two types of enrichment functions, one used to present the discontinuities across the crack and another used to present the singularity of stress field close to the crack tip, are incorporated into displacement approximation. Stress intensity factors can be calculated using J-integral method. Zarrinzadeh [81] presented a 3D degenerated shell model with XFEM method to study the fatigue crack growth of materials. The crack growth against the number of applied cycles of load and the fatigue crack trajectories were found in good agreement with the experimental findings. The strength of structure is increased by wrapping the cracked pipe with glass/epoxy polymer composite. Farhad [82] coupled VCCT and XFEM method to simulate mode I fatigue delamination propagation in composite. The results showed that use of XFEM reduces the computational cost significantly. The initial time increment has no influence on the XFEM simulation results. The running time is less than the VCCT method. According to the results, the number of cycles required for delamination onset under force control is less than the displacement control loading. The decrease of reaction force of load points is faster within the cycle number of 50,000.

The cohesive zone model (CZM) approach is another widely used method in modelling crack propagation in interface. CZM has the capability to model the delamination onset and propagation without defining the initial defect, unlike the fracture mechanics method [2,55,83]. In the VCCT method, complex algorithms are mandatory to monitor the crack tip numerically; it permits crack growth by removing constraints on identical nodes. The projected crack propagation path must be clear in advance. To overcome these limitations, an alternative method is the implementation of a cohesive zone degradation law into interface elements [2,3,84,85]. The CZM methodology has many benefits in dealing with complex geometries without remeshing and fatigue crack propagation modelling in finite element framework [86]. The CZM describes a softening relationship between tractions and relative displacements of each pair of adjacent nodes. Figure 16 shows the schematic diagram of the cohesive law. Before damage starts to grow, the elastic behavior is governed by the elastic stiffness as follows:(5)σ=kw,
where k represents the interfacial stiffness.

After reaching the maximum interfacial stress, the damage occurs. In fatigue loading conditions, there are two possible approaches of modelling interface element damage accumulation [87]. The first one tracks loading/unloading and degradation of interface stiffness on a cycle-by-cycle basis. This approach can also be advantageous for ductile material under low-cycle fatigue applications. The second method applies a loading envelope approach, where the numerically applied load remains persistent at the highest value of the cyclic load, and after each model time step the degradation of interface element occurs based on a distinct number of the following elapsed cycles. This approach eliminates the necessity to explicitly simulate every distinct loading cycle that offers high computational effectivity. This method is more appropriate for high cycle fatigue applications, which may involve additional 10 × 6 cycles.

In cycle-by-cycle loading/unloading fatigue damage formulation, tracking the accumulation of permanent damage and simulation of progressive damage in the softening region of the CZM is achieved by
(6)σ=(1−d)kw,
where d represents the damage parameter ranging between zero (undamaged case, $w_{i}<w_{0}$ in Figure 16) and one (complete failure, $w_{i}=w_{u}$ in Figure 17). The value of d is updated in each increment and given by
(7)$$d=\frac{w_{u}\left(w_{i}-w_{0}\right)}{w_{i}\left(w_{u}-w_{0}\right)},$$
where $w_{u}$ and $w_{0}$ represent the ultimate relative displacement leading to complete failure at the integration point and the relative displacement conforming to damage onset, respectively. $w_{i}$ is the relative displacement of integration point at increment *i*. A similarity was found between the CZM and experimental results by Moura [88]. When compared with load ratio of 0.2 (load ratio $R=P_{min}/P_{max})$, the final fracture with load ratio of 0.4 occurred later due to the less sever fatigue loading in the earliest 50,000 cycles. Occurrence of drastic failure increased from 90,000 cycles at load ratio of 0.2 to 200,000 cycles at load ratio of 0.4. The increased load ratio lead to longer fatigue life.

**Figure 16 polymers-14-04558-f016:**
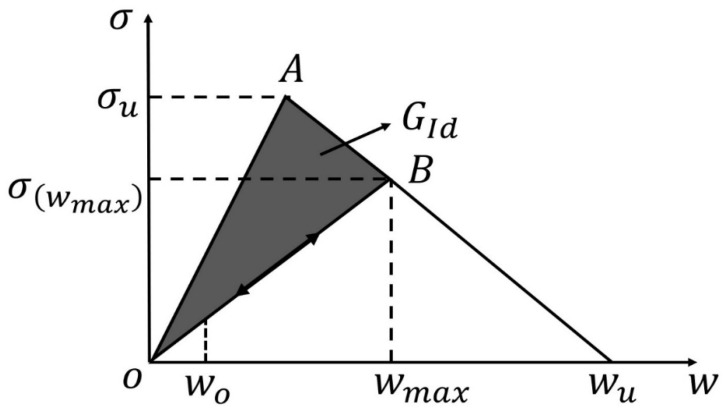
Schematic representation of the cohesive law [89].

For CZM defined with loading envelope strategy, the softening law was regarded as a function of number of cycles. For each model time-step, the fatigue damage parameter is updated using
(8)$$d=d_{f,old}+f\delta t\frac{\partial d_{f}}{\partial N},$$
where f is the user-defined number of cycles per second, $d_{f}$ is the fatigue parameter, δ represents the strain at integration point and t represents the step time. Harper [90] established a CZM defined with loading envelope strategy. A mode I maximum interface stress of 15 MPa and 30 MPa was applied to check the accuracy of fatigue formulation for different lengths of cohesive zone and different interfacial properties. The results show that decreasing the maximum interfacial stress increases cohesive zone length. The established cohesive zone length was 1.2 mm and 2.4 mm, respectively. Close relationship occurs between both sets of numerical results and the theoretic Pairs curve. This indicates that CZM is not meaningfully affected by the number of elements within cohesive zone. Crack propagating rate had been accurately predicted under fatigue loading.

F. Teimouri [91] extended the Turon et al. and Kawashita-Hallett’s CZM fatigue damage models with a trilinear cohesive law to simulate mode I fatigue delamination in the composites undergoing large-scale fibre bridging. To simulate 3D DCB specimen behaviour under high loading cycles, a user defined element (UEL) subroutine was applied to the Abaqus software. By assessment with the existing data from the literature, the trilinear cohesive laws presented well precise results in the fibre bridging simulations. The suitable tuning parameters in each extended models were proposed to get the best numerical results. The results exhibited that fatigue crack growth (FCG) rate projected by the extended model depends on the fatigue cohesive zone lengths, quasi-static load, and the increment of cycles.

Skvortsov used shell elements instead of 2D elements or solid elements to model fatigue delamination propagation in composite laminates. Loading envelope strategy combined with algorithm of fatigue damage is defined in a CZM model. Delamination length and fatigue response of composite under fatigue loading were simulated. The proposed approach was found efficient and sufficiently accurate. The fatigue life of the examined specimen was accurately predicted. The delamination length dramatically increased after 60,000 cycles. The fatigue and quasi-static response of composite were obtained with proposed model.

The numerical simulation method has been increasingly accepted as a valuable tool for predicting the damage initiation and propagation. It has been an essential technique in fatigue life prediction, failure mechanisms investigation and fatigue strength optimization of composite.

## 6. Testing Technique

There are many testing techniques (destructive and non-destructive) and specimen configurations to determine the mode I fatigue delamination of laminated composites such as the double cantilever beam test (DCB) [36,37,41,42,83,92], round-robin mode I fatigue fracture tests, digital image correlation (DIC), X-ray radiography scan [41], ultrasonics thermography and acoustic emission detector [45]. However, the limited resolution of conventional non-destructive testing techniques makes them inadequate for sensing the initiation and propagation of short fatigue cracks.

DCB configuration as shown in Figure 17 is most common in the mode I fatigue tests. The ASTM D6115-97 is the standard method for mode I fatigue characterization of composite laminates [93]. This standard also recommends the DCB specimen for fatigue delamination test. Al-Khudairi et al. [94] characterized the mode I cyclic delamination threshold onset life of FPR laminated composites by following ASTM D6115-97 standard method. Another researcher followed the same standard to find the cyclic delamination progression onset of woven/braided fibre composites [76].

The DIC is one of the most widely used techniques to determine the mode I fatigue delamination in composites. Zhu et al. [92] used the DIC-based technique to analyse the initiation and growth of mode I fatigue delamination in composite materials. The DCB specimen (as shown in Figure 17) was used to analyse the structure of the composite under fatigue loading [2]. The DIC was used to observe the displacement field on the edge of the DCB sample, from which the crack tip position was examined as a place where opening displacement converges to zero. Both crack initiation and propagation under fatigue loading were determined using this technique. In contrast to the compliance-based technique suggested in the ASTM standard ASTM D5528 for quasi-static and ASTM D6115 for fatigue loading, the DIC-based technique (which depends on the specific crack tip location) provides the conservative value for fatigue initiation. The DIC-supported technique decreases the quantity of worker and structure-dependent failures can identify the delamination in real-time under fatigue loading [92].

Maillet et al. studied the impact of frequency on cyclic mode I fatigue delamination growth and compared it with standard tests accomplished at 10 Hz frequency. The vibration resonance machine has been used to measure the mode I fatigue delamination rate of composite [52]. A dynamic process used mass spring-based system that was equipped to test delamination propagation under applied load and stopped the test under reproducible situations. That method enabled the fatigue analysis to be performed at 100 Hz, decreasing the time required by the tenth factor while avoiding harmful heat generation in the material. When the loading frequency was increased, the resistance to delamination propagation was decreased as shown in Figure 18, and SEM images of the fractured surfaces are shown in Figure 19.

On the other hand, Brunner et al. [14] found the fatigue failure of composite using the double cantilever beam method from the quasi-static mode I delamination resistance experiment. The testing method, calculation and data collection have been established with a concentration on industrial testability. Several research criteria have been modified to observe how they affect the outcome. Two methods of calculating the delamination length were compared. Visible delamination length estimation and effective delamination length (based on compliance-based method and single modulus of electricity accurate) both produce appropriate acceptance (Figure 20). It makes sense that it might be beneficial to continue developing the research method to include automatic data collection and detection.

## 7. Methods to Improve Mode I Fatigue Delamination

### 7.1. Matrix Toughening/Particles Interlayering

Another method known as interlayer of interleaf is the use of toughened resin system to improve the interlaminar fatigue of carbon fibre reinforced composites. This system involves adding thermoplastic or thermoset particles to the resin system [16,30]. Interleaving concept to enhance the fatigue crack propagation in metallic structures has been used in the aircraft industry for a long time. Martin and Murri [18] investigated mode I fatigue delamination initiation in AS4/PEEK composites. According to their findings, thermoplastics did not increase the mode I cyclic delamination resistance significantly. Hojo et al. achieved similar result with interleaved self-epoxy carbon/epoxy laminated composites [16].

Stevanovic et al. [95] used interlayer toughening approach to increase the mode I delamination resistance under dynamic loading of glass/vinyl ester composites. Several vinylester (VE)/poly(acrylonitrile-butadiene-styrene) ABS mixtures were used as interlayer between plies of composite. Hojo et al. [55] and Sato et al. [96] also used polyamide particles as thermoplastic resin interlayer material to increase cyclic crack growth resistance of carbon/epoxy (T800H/3900-2) laminated composite under mode I loading. The threshold value of interlayered composite was 3.3–3.5 times higher than that of reference laminate [55]. The crack growth resistance increased due to the crack growth propagation path from toughened interlayer region initially, but as the crack path transitioned to untoughened region, cyclic crack delamination resistance also decreased but was higher than the base composite. Figure 21 shows the schematic presentation of crack path transition from toughened (having polyamide particle) to untoughened region during mode I fatigue loading (b) [55]. This interlayered composite has become a choice in the Boeing 777 construction because of its higher impact damage resistance and interlaminar resistance [97].

However, particle interleaved composites have some drawbacks such as decrease in in-plan stiffness and strength (15–20%)and increase in laminate thickness (~20%) and potential decrease of $T_{g}$ [30]. Electrospun polymeric nanofibre fabric interleaving is reported as a potential solution to reduce or eliminate the above-mentioned problems.

### 7.2. Nanotubes/Nanofibres Interleaving

Structural properties such as high modulus, high strength and good creep resistance of fibre reinforced polymeric composites can be enhanced by infusing nanoparticles. Graphene, carbon nanofibres (CNFs), spherical nanoparticles, carbon nanotubes (CNTs)are all outstanding candidates for reinforcement which are dispersed in matrices in smaller amount and present improvements in properties of polymeric materials [21].

The trend of cellulose nanofibres (CeNFs) has much increased due to their higher mechanical properties, excellent sustainability, production, and demand for the growth of composites [98,99,100,101]. The demand for CeNFs for the growth of composites is quite a different field in research. CeNFs can be used as a reinforcement to increase the mechanical, thermal and biodegradation performance of the composites [101,102]. Shivakumar et al. [30] observed a significant effect of nanofibres interleaving to delay the delamination onset for carbon/epoxy composite under mode I fatigue loading condition. Brugo et al. [96] also found that incorporation of Nylon 66nanofibres to carbon/epoxy woven laminates increased the strain energy required for delamination initiation and threshold energy release rate increased by 90% compared to the plain composite shown in Figure 22.

Ladani et al. [103] found that by the addition of CNFs into carbon-reinforced polymer composites (CRPs), the cyclic fatigue resistance was enhanced, and debonding behaviour in adhesive-bonded structure was recognisable. They studied the influence of concentration and the orientation of the CNFs in the epoxy adhesive layer between two CFRP substrates. It was seen that the mode I fatigue resistance of the adhesive layer was increased by increasing the concentration of randomly oriented CNFs. When the carbon nanofibers were arranged perpendicular to the plane of the joint, the fatigue resistance was more increased [103].

Romhany et al. [104] investigated the effect of carbon nanotube addition on the cyclic mode I interlaminar behaviour of carbon fibre reinforced composites. The authors used acoustic emission (AE) technique to track the interlaminar fatigue crack. They reported a decrease in crack propagation rate by 69% as well as a significant improvement in fatigue life. Christopher et al. [105] reported a significant increase in cyclic SERR of glass/epoxy composites under mode I fatigue loading by addition of small quantity of multi-walled CNTs to the matrix.

Many studies [103,106,107,108,109,110] in the literature reported the improvement in fatigue resistance of composites by incorporation of nano-clays, CNTs and CNFs; however, there is no consensus on the effect of content/amount of these external particles on fatigue resistance.

### 7.3. Z-Pinning

Z-pinning or Z-binder yarns are well-known approaches for improving the fatigue delamination resistance of fibre reinforced composites [26]. Z-direction reinforcement can interrupt the delamination. The fatigue life of composite improves due to the decrease of the interfacial stress concentration which transforms the damage mechanism from delamination to fibre breakage. Z-pins are needle-shaped, round strips with a diameter of about 0.1 to 1mm that are introduced into the composite thickness before curing (Figure 23). Titanium, steel, or carbon fibre reinforced composites are commonly used for these pins because they have high strength and stiffness.

J. Hoffmann and G. Scharr [111] analysed the effect of various geometric shape pins on composites’ fatigue resistance with two different ply-stacking sequences. According to the Paris curves, the rectangular z-pins were more efficient than the circular z-pins in growing delamination resistance of quasi-isotropic composites. In unidirectional composites, when the cross-section of the pins was changed, it slightly affected the mode I fatigue delamination resistance of the z-pins composite.

Similarly, z-pins were used to analyse the mode I fatigue resistance of carbon–epoxy laminated composite by incorporating them in through-thickness direction [112]. Zhang et al. [113] explored the effect of bridging forces caused by inserting z-pins on fatigue delamination behaviour of carbon/epoxy laminated composite; z-pins of two different sizesand two different volume contents were inserted in through-thickness direction of carbon–epoxy laminates. The bridging forces and debonding decreased by decreasing the diameter of z-pins and vice versa.

Pingkarawat and A.P. Mouritz [114] analysed that upon the increase in the content of z-pins up to limiting concentration by volume, the delamination crack rate was suddenly decreased under the mode I cyclic load. It was also observed that by increasing the length of the z-pin, or when the diameter of the z-pin was decreased, the rate of delamination also decreased (Figure 24). Beyond a critical point, increasing the z-pin length does not bring any improvement to fatigue resistance due to transition of the cyclic mode I delamination development from single to multiple cracks as shown in Figure 25 [115].

Denis et al. [34] reported a study on mode I fracture and fatigue delamination behaviour of z-pin reinforced carbon/fibre epoxy laminated composites. They compared the unpinned composite samples with carbon fibre z-pins containing composites samples. Z-pinning resulted in a slowdown in fatigue crack propagation, and at increased areal density of z-pins (from 2–4%) the maximum fatigue load increased from 170 N to 250 N. Hojo et al. [116] examined the mode I fatigue delamination in carbon/epoxy cross-ply composite with z-anchor reinforcing. Using resin film injection, the composites were manufactured with dry z-anchor reinforcements. The double cantilever beam method was used in the mode I fatigue delamination experiment. They concluded that z-anchor reinforced composites have fatigue threshold values of 3.4 to 5 times greater than the composite without z-anchor reinforcement.

Another study [117] reported the effect of z-pins on carbon fibre, steel, copper and titanium on mode I fatigue delamination resistance. Contrary to many studies in which authors investigated the influence of z-pins content and size, this study [117] investigated the effect of z-pin material on fatigue delamination. The metal (steel and copper) and carbon fibre z-pins were found to be more effective at increasing the mode I fatigue delamination resistance as compared to copper and titanium, and this correlates with their tensile fatigue strength.

Z-pins made of low fatigue resistant material such as copper collapse quicker under mode I fatigue loading than z-pins comprised of high fatigue resistant materials such as carbon fibre, and consequently this reduces their capability to enhance the cyclic delamination properties. In a similar study, Abbasi et al. [51] quantified and compared the interlaminar fatigue resistance of metal z-filaments (thin steel and copper wires) with carbon fibre z-filaments under mode I fatigue loading. Z filaments were used in through-thickness direction of carbon fabric preform at weaving stage, then saturated with resin to fabricate 3D woven composite. Fatigue tests were performed using DCB specimen. The reported results show that metal z-filaments were found to be less effective in increasing the mode I fatigue than carbon filaments. This was attributed to the highest fatigue strength of carbon filaments, which produced higher bridging traction loads along crack path than metallic filaments as shown in Figure 26. Among metallics, filament steel performed better in comparison to copper.

Therefore, we can summarize that mode I fatigue properties of z-pinned reinforced composites can be tailored by the right choice of z-pins content, size (length and diameter) and material. Table 2 describes the effect of matrix toughening, interleaving and z-pinning on improving mode I fatigue performance of composite relative to base composites.

The polymeric composite interfaces can be changed with thermoplastic layers or PA66 fibres to enhance their crack growth resistance and fatigue life. For example, CFRP interface was modified by addition ofPA66 nanofibres of approximately 520 nm diameter, which decreased the crack propagation rate up to 30 times [118]. The cracks propagated in different planes (thickness and width directions) when bounced between carbon fibres and the toughened PA66 nano-modified layer, requiring higher energy for further growth. Shivakumar et al. [30] also reported similar results for PA66 nano-modified specimens, where a substantial delay in delamination onset was seen under fatigue loading of specimens. The axial fatigue life (compression, tension dominated) of carbon nanofibres can be increased by between 150% and 670% by interleaving as result of increased damage protective effect of the nanofibres and interface density.

**Table 2 polymers-14-04558-t002:** Effect of matrix toughening, interleaving and z-pinning on mode I fatigue performance of composite.

Fibre/Resin	Functional Material	Property	Improvement	Reference
Carbon/epoxy	Inomer interleaf	Fatigue threshold	3 times better	[55]
Carbon/epoxy	Polyamide nanofibres	Fatigue threshold	90 %improvement	[96]
Carbon/epoxy	50 µm epoxy interleaf	Fatigue threshold	No improvement	[16]
Carbon/epoxy	0.7 weight% Carbon nano fibres	Fatigue threshold	6 times>	[103]
Carbon/epoxy	0.3 weight% multi walled carbon nanotube	cycles to failure	3.8 times>	[104]
Glass/epoxy	1 weight% carbon nanotube	Fatigue threshold	Slight improvement	[105]
Glass/epoxy	2 volume % z pins by	Fatigue threshold	15 times>	[112]
Glass/epoxy	2 volume % z pins by	Strain energy release rate range	13–15 times>	[114]

## 8. Summary and Future Directions

Exceptional properties with low weight make composite materials ideal for the era. However, several factors influence composite materials’ service life, among which fatigue is a prominent phenomenon. Composite material parameters are viable in governing mode I fatigue. Textile preform architectural variations govern mode I fatigue behavioural differences among composite materials. Multidirectional laminated composites perform better in mode I fatigue than unidirectional reinforced composites. Z-yarns in through-thickness direction significantly enhance mode I fatigue behaviour of 3D woven reinforced composites. Though 3D reinforced composites have excellent mode I fatigue performance, very limited studies reported their fatigue behaviour under mode I cycling loading.

Both compression and tensile failure modes are involved in mode I fatigue loading. Reinforcement fibre failure occurs due to imbalanced arrangements and debonding in compression and tensile modes, respectively. In contrast, matrix fails due to inter-fibre fractures initiated by interlaminar stresses. Although environmental factors such as temperature and aging have a pronounced effect on fatigue life of composites, nevertheless very limited research is found on the effect of environmental factors on mode I fatigue. A small number of studies on mode I fatigue reports the effect of temperature, i.e., temperature is influential in defining fatigue life. The effect of aging on mode I fatigue delamination resistance is not currently captured. Numerical studies also present models for predicting the mode I fatigue life of composite materials.

Testing techniques, including cantilever beam tests, digital image correlations, X-ray tomography, and acoustic emissions detections are used to predetermine fatigue behaviours of composite materials’ service life. In a nutshell, tremendous work has been done in the literature regarding mode I fatigue behaviours of composite materials, opening several new ways to explore further. Different techniques employed for mode I fatigue life enhancement are fruitful, including resin toughening, z-pins insertion and nano modification of composites using carbon nanotubes, cellulose nanofibres, etc. However, there are discrepancies on the effect by volume or weight content of these nanomaterials and their working mechanisms that need further attention of researchers.

## Figures and Tables

**Figure 1 polymers-14-04558-f001:**
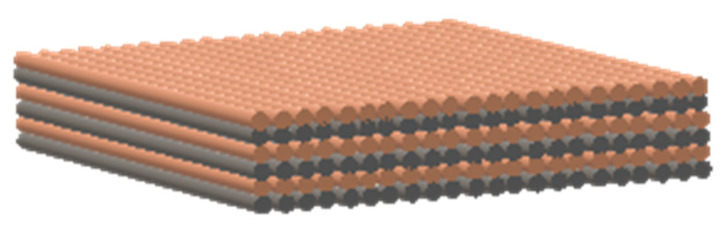
Unidirectional structure.

**Figure 2 polymers-14-04558-f002:**
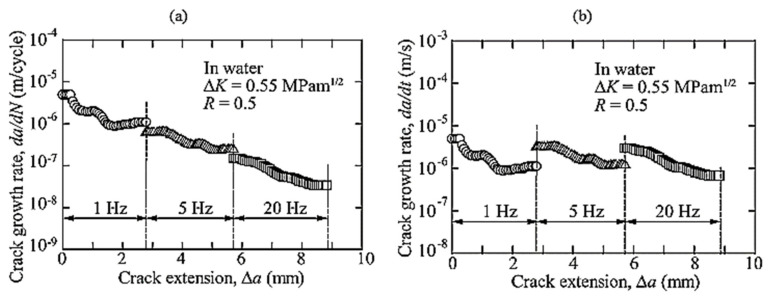
Impact of loading frequency on mode 1 fatigue cracking of M4J/2500 laminates in water (**a**) cycle-dependent cracking rate, (**b**) time-dependent cracking rate [35].

**Figure 3 polymers-14-04558-f003:**
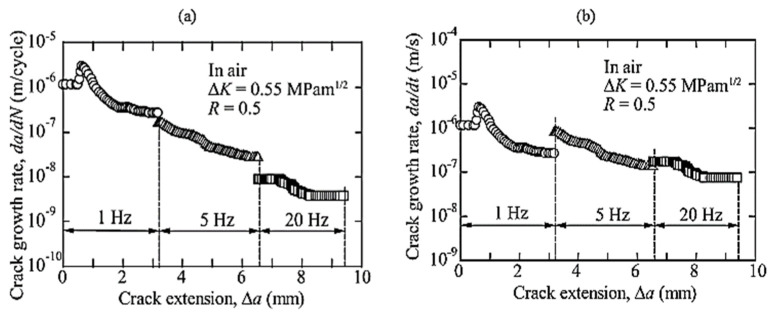
Impact of loading on mode 1 cracking of M4J/2500 laminates in air (**a**) cycle-dependent cracking rate, (**b**) time-dependent cracking rate [35].

**Figure 4 polymers-14-04558-f004:**
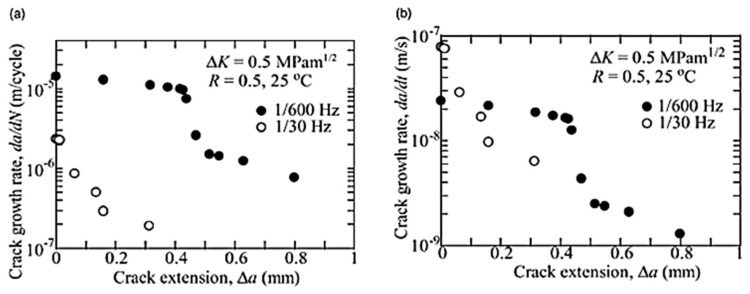
Impact of loading on mode 1 fatigue cracking of T300/3601laminates in water (**a**) cycle-dependent cracking rate, (**b**) time-dependent cracking rate [35].

**Figure 5 polymers-14-04558-f005:**
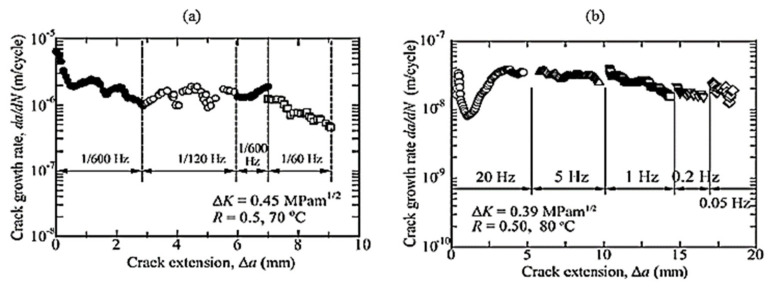
Impact of loading frequency on mode I fatigue of T300/3601 laminates in air (**a**) 1/600–160 Hz, (**b**) 0.05–20 Hz [35].

**Figure 6 polymers-14-04558-f006:**
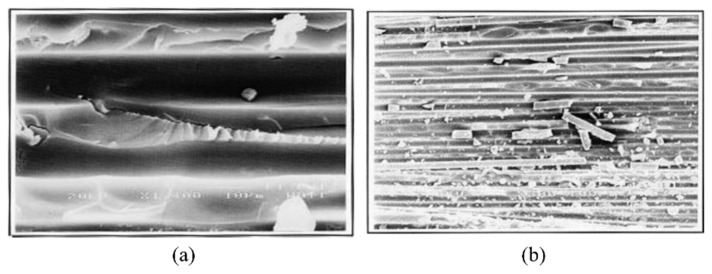
SEM image of fractured surface after mode I fatigue loading: (**a**) path cleavage (**b**) fibre and matrix fragments [36].

**Figure 7 polymers-14-04558-f007:**
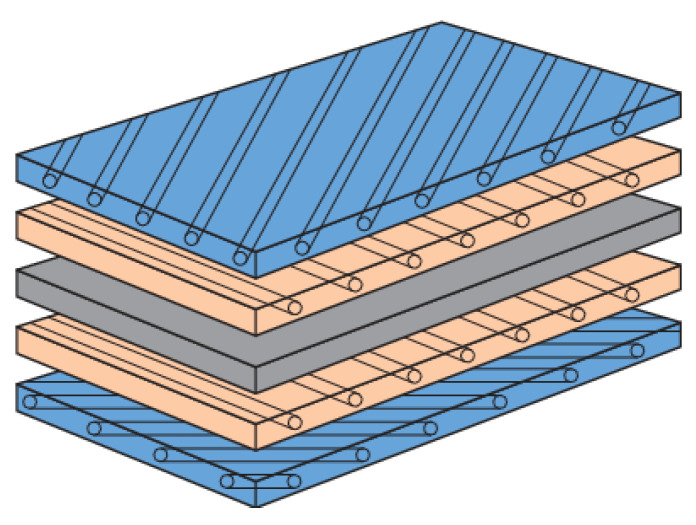
Multidirectional laminated composite structure.

**Figure 8 polymers-14-04558-f008:**
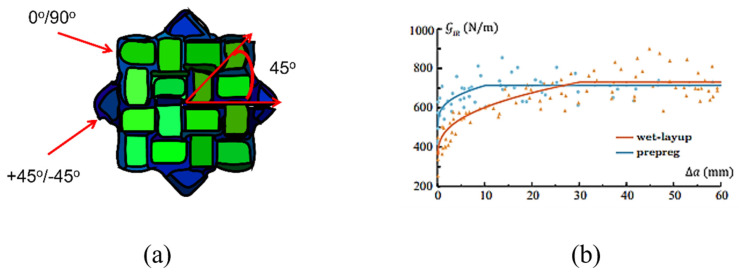
Stacking sequence of multidirectional laminated composite (woven prepreg) (**a**); Fatigue delamination resistance curves of prepreg and wet-layup (**b**) [44].

**Figure 9 polymers-14-04558-f009:**
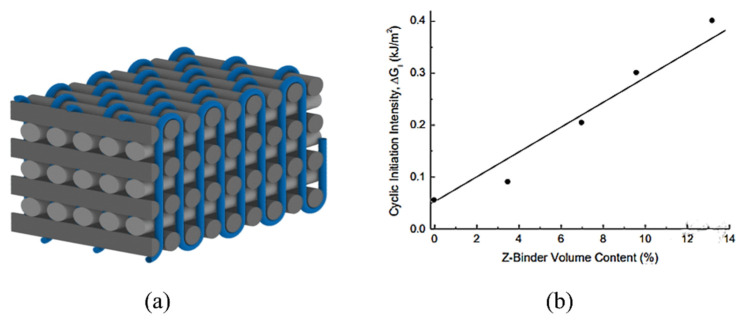
3D Woven reinforcement (**a**) and influence of z-binder yarn volume on the cyclic stress intensity factor (**b**) [50].

**Figure 10 polymers-14-04558-f010:**
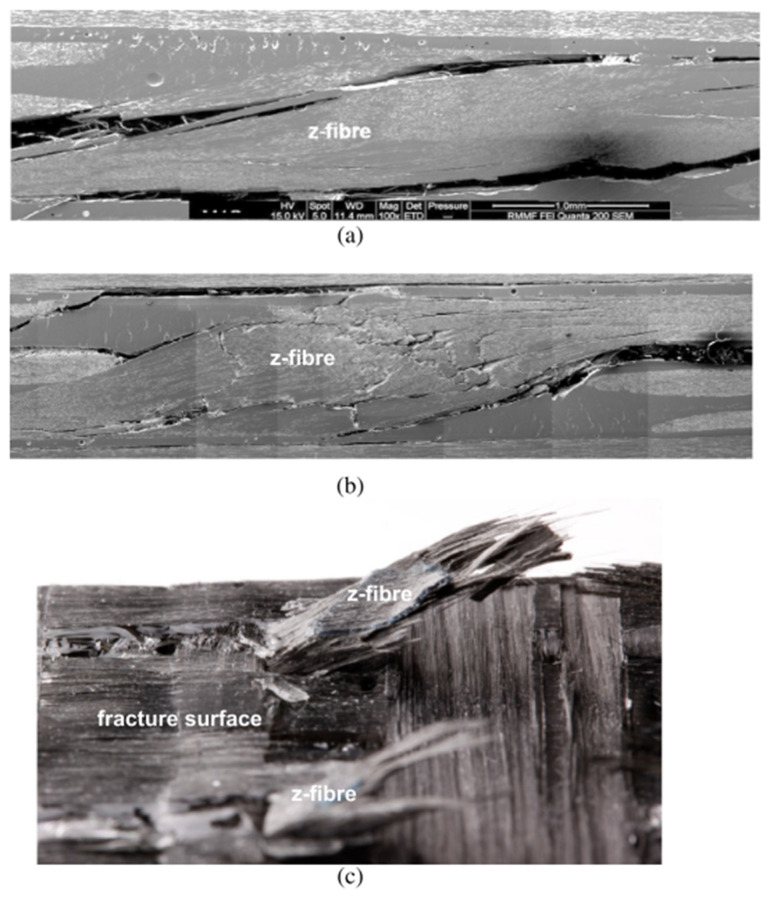
Influence of z-binder yarn on cracking in 3D woven composites under mode 1 fatigue loading: (**a**) interfacial cracking between composite and z-binder yarn, (**b**) interfacial cracking of fibre–matrix and breaking of fibre within z-binders, (**c**) z-binders pull-out failure [50].

**Figure 11 polymers-14-04558-f011:**
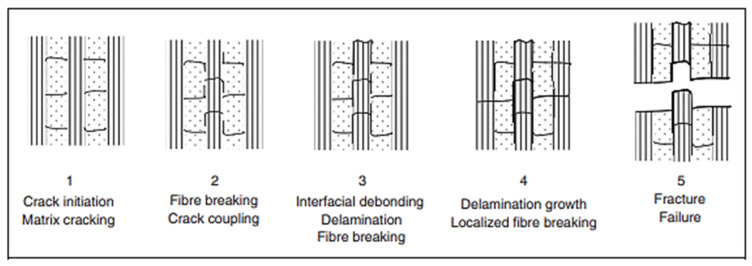
Damage formation in cross-ply laminate during tension–tension fatigue.

**Figure 12 polymers-14-04558-f012:**
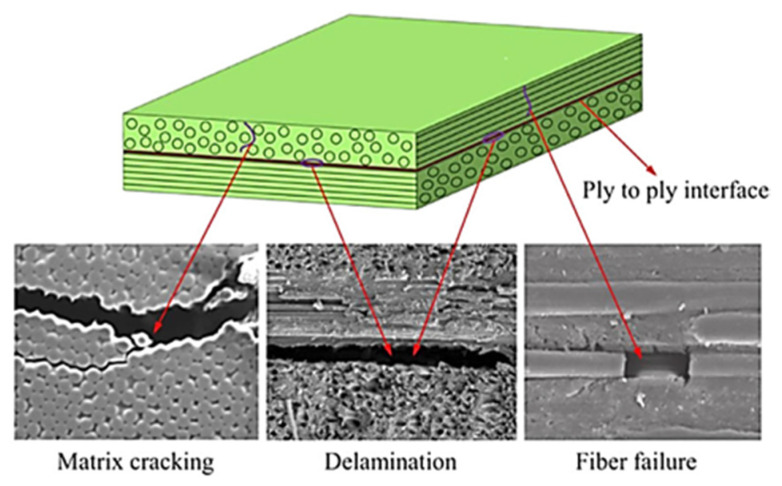
Fibre failure and matrix failure within mode I fatigue loading.

**Figure 13 polymers-14-04558-f013:**
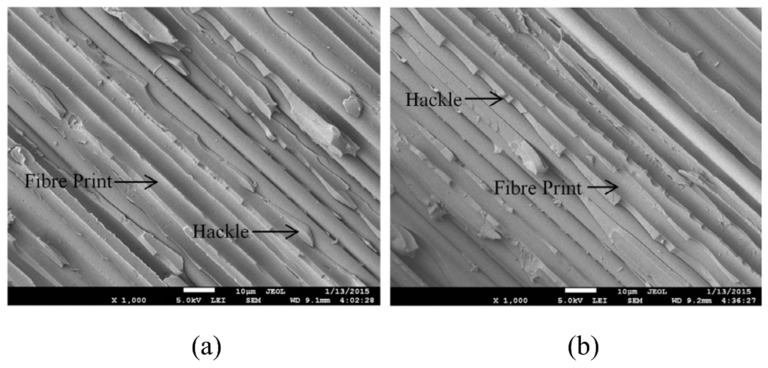
SEM images of fractured surfaces under mode I fatigue loading (**a**) at short crack length (**b**) at long crack length.

**Figure 14 polymers-14-04558-f014:**
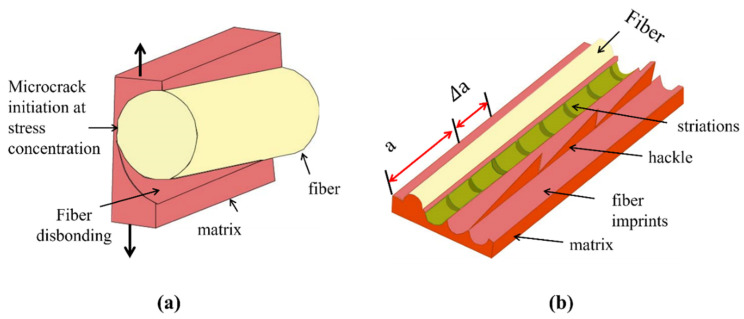
Schematic diagram of (**a**) microcrack initiation in matrix and fibre disbanding (**b**) hackles and striations per unit length [31].

**Figure 15 polymers-14-04558-f015:**
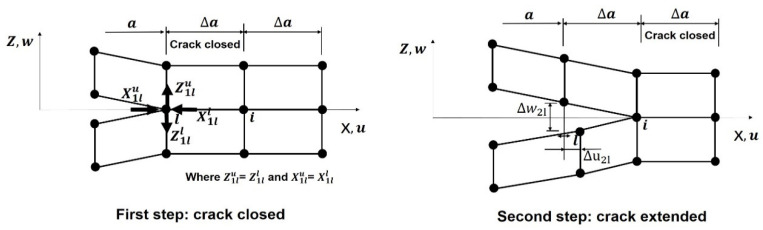
Two-step crack closure method [77].

**Figure 17 polymers-14-04558-f017:**
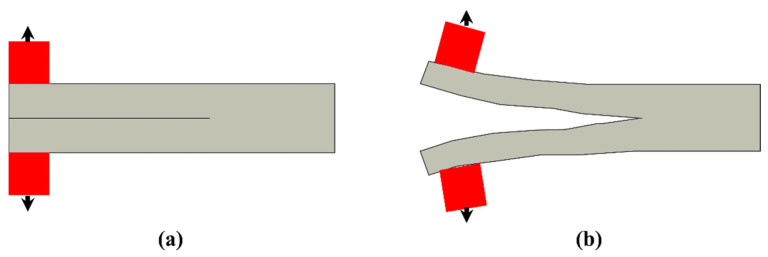
Sketch of DCB sample (**a**) undeformed (**b**) deformed.

**Figure 18 polymers-14-04558-f018:**
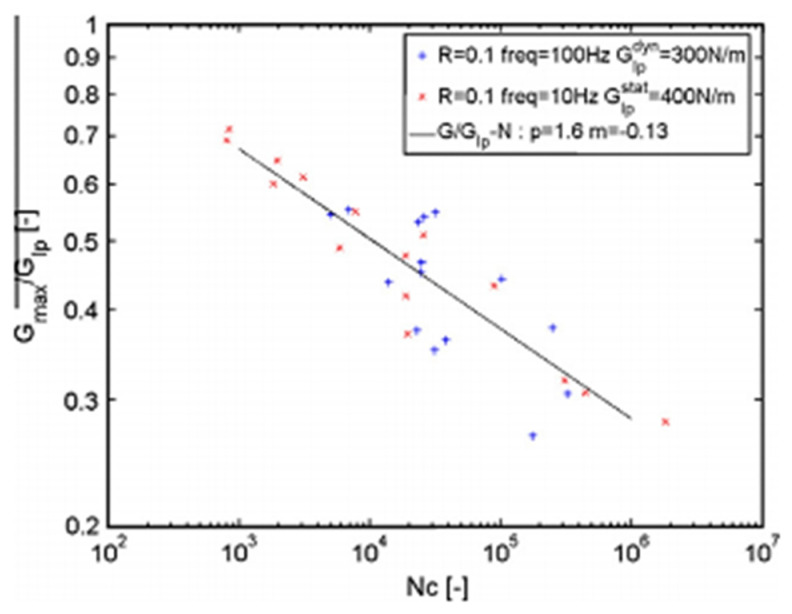
G–N propagation curve on G_Ip_ for 10 Hz and 100 Hz [52].

**Figure 19 polymers-14-04558-f019:**
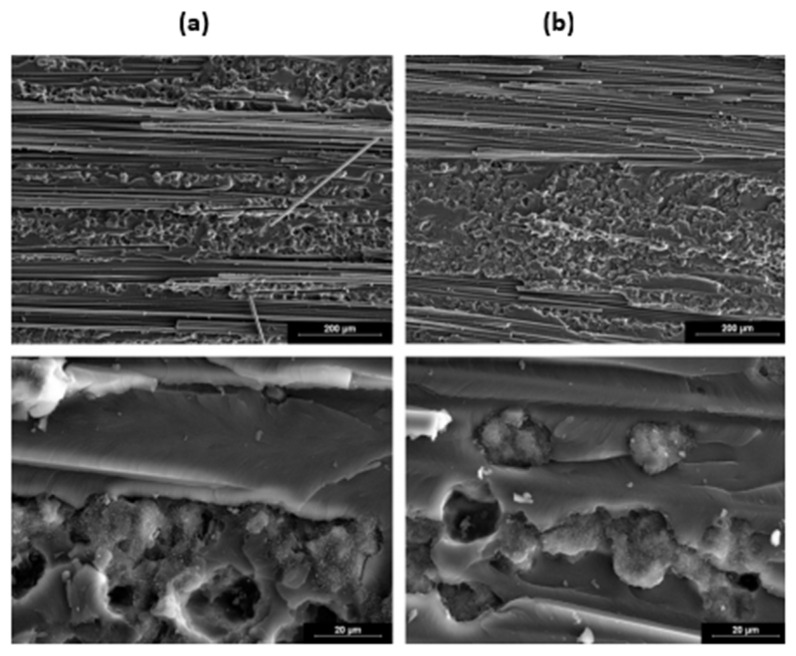
Fracture surface (**a**) for 10 Hz (**b**) for 100 Hz [52].

**Figure 20 polymers-14-04558-f020:**
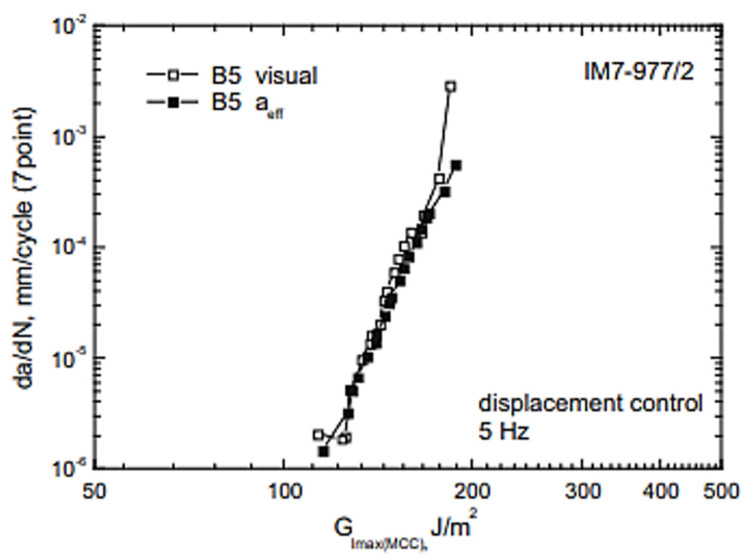
Comparison of visual versus effective delamination length measurement [14].

**Figure 21 polymers-14-04558-f021:**
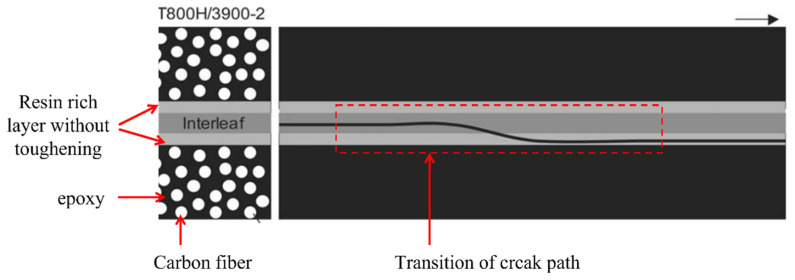
Schematic presentation of crack path under mode I cyclic loading.

**Figure 22 polymers-14-04558-f022:**
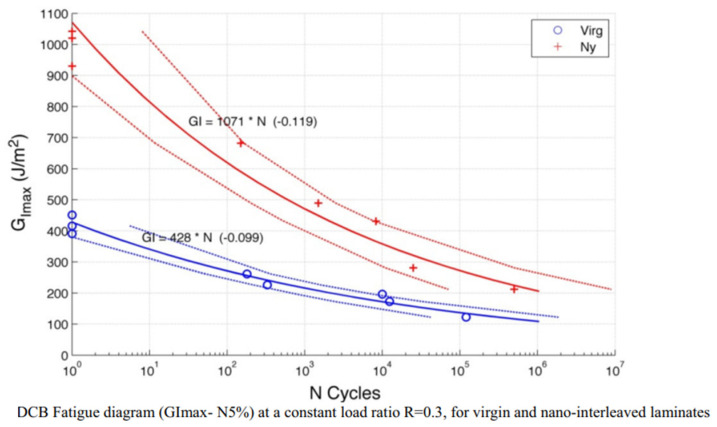
Strain energy release rate versus No. of cycles for nylon 66 nanofibre interleaved and virgin composite.

**Figure 23 polymers-14-04558-f023:**
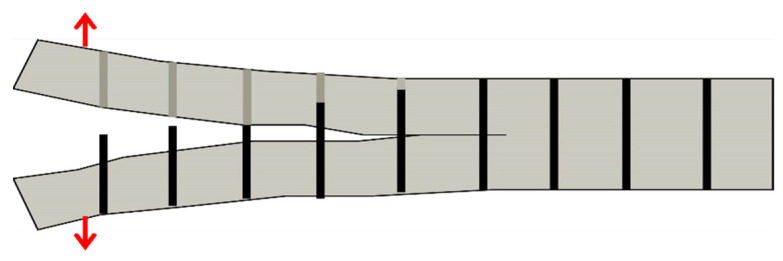
Schematic of z-pinned specimen with a steady state crack growth.

**Figure 24 polymers-14-04558-f024:**
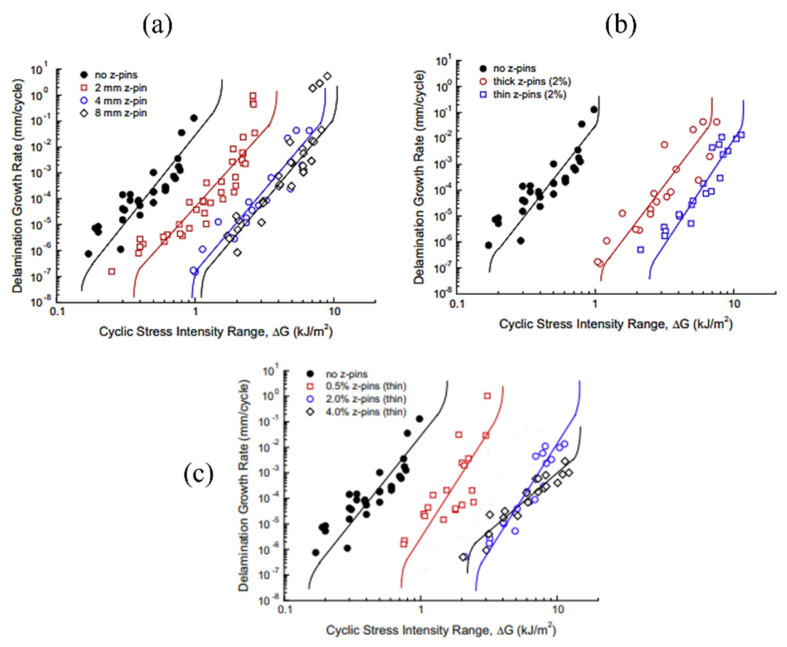
Effect of z-pin dimensions and volume on fatigue crack growth rate (**a**) Effect of length (**b**) Effect of diameter (**c**) Effect of volume [112].

**Figure 25 polymers-14-04558-f025:**
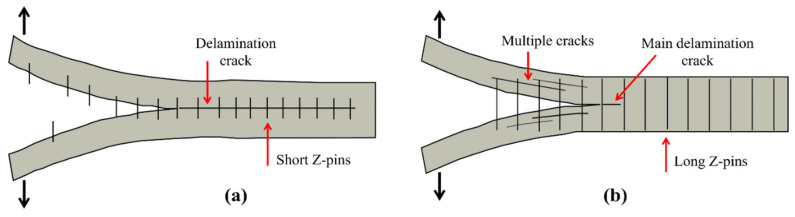
Schematic presentation of mode I fatigue delamination behaviour of composite with z-pins (**a**) short z-pins (**b**) long z-pins.

**Figure 26 polymers-14-04558-f026:**
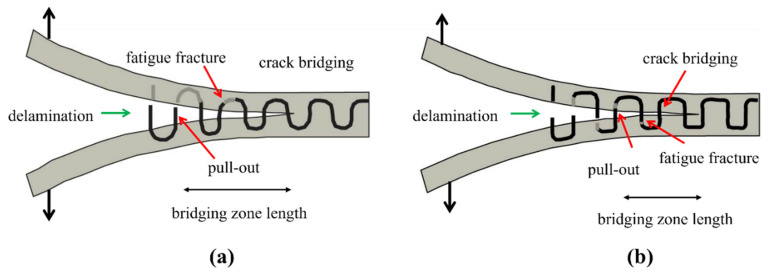
Schematic diagram showing the sideview of DCB specimen subjected to mode I fatigue (**a**) steel z-filaments (**b**) carbon fibrez- filaments.

**Table 1 polymers-14-04558-t001:** Fatigue performance of glass/carbon fibre reinforced polymeric composites.

Fibre/Matrix	Structure	Test/Specimen Type	Cyclic Stress Intensity Factor Range/Strain Energy Release Rate Range	Reference
Glass-Carbon/epoxy	Multidirectional	DCB	51.8 N/m 139.5 N/m	[43]
T700/M21	Unidirectional	ASTMD 6115/DCB	300 N/m	[52]
Carbon/epoxy (3501-6) prepreg	Unidirectional	ASTM 6115-97/DCB	101.99 J/m^2^	[53,54]
Carbon/epoxy (8552) prepreg	Unidirectional	ASTM 6115-97	124.50 J/m^2^	[53,54]
Carbon/epoxy-T800H/3900-2	Unidirectional	DCB	400 J/m^2^	[55]
Carbon/epoxy (AW 196), Carbon/epoxy (8552)	Unidirectional	ASTM D6115/DCB	151–26.6 J/m^2^	[56]
Carbon/epoxy	3D woven	DCB	0.5–0.4 kJ/m^2^	[50]
Glass/epoxy	Laminated	DCB	7–10 J/m^2^	[8]
Carbon-Glass/epoxy Prepreg	Multidirectional	ASTM D6115-97/DCB	30.7, 56.1, 96.0, 331.0 N/m	[44]
Carbon-Glass/epoxy Wet layup	Multidirectional	ASTM D6115-97/DCB	51.5, 141.0	[44]

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
