# Peer review of "Mode I Fatigue of Fibre Reinforced Polymeric Composites: A Review"

_polymers, 2022, doi:10.3390/polym14214558_

Round 1
Reviewer 1 Report
The paper is structured properly (MATERIAL AND METHODS, DISCUSSION, CONCLUSIONS, REFERENCES, etc.). The ABSTRACT section is well–structured. The INTRODUCTION section provide the necessary background information, although it is quite long this section. In its entirety, the article is quite long (41 pages). This material is not a simple review, it is a book chapter.
The METHODOLOGY (METHODS) is relatively well described. Also this section is quite long and would require a better synthesis and compaction of the ideas exposed.
The body of paper describe the important RESULTS of the research. The CONCLUSION section succinctly summarize the major points of the paper. The list of REFERENCES is long and relatively well chosen.
A shortening of this article would be required. It would be desirable for authors to have a higher synthesis capacity, for an article (scientific paper).
Author Response
Reviewer 1:
Comments and suggestions
The paper is structured properly (MATERIAL AND METHODS, DISCUSSION, CONCLUSIONS, REFERENCES, etc.). The ABSTRACT section is well–structured. The INTRODUCTION section provide the necessary background information, although it is quite long this section. In its entirety, the article is quite long (41 pages). This material is not a simple review, it is a book chapter.
Reply: we are very thankful to the reviewer for these valuable comments. We make the introduction section short.
The METHODOLOGY (METHODS) is relatively well described. Also, this section is quite long and would require a better synthesis and compaction of the ideas exposed.
Reply: Actually, in this review articles we tried to summarize all testing methods of Mode I fatigue delamination as well as we have given methods for improvement of fatigue delamination in order to help researchers and designers to make structures with higher fatigue resistance.
The body of paper describe the important RESULTS of the research. The CONCLUSION section succinctly summarizes the major points of the paper. The list of REFERENCES is long and relatively well chosen.
Reply: Thank you for this comment.
A shortening of this article would be required. It would be desirable for authors to have a higher synthesis capacity, for an article (scientific paper).
Reply: As per reviewer comment we have tried our best to make this article short.
Reviewer 2 Report
The article is an extensive material on polymer composites. The authors analyze the extensive literature. concerning both the theory and practice related to the production of composites and their properties. The literature contains 113 items. Despite the great value of the work, I would like to note that a similar study has already been done, there is also a book study. My doubts are whether the journal was correctly selected for the publication of this type of article. Here my comments:
1. What is the thesis of the work, what is the specific goal.
2. The topic is not entirely original, but it presents a great substantive value related to the development of composites. The main goal should be edited and the research carried out more emphasized, so that the reader can clearly know what has been achieved in the work.
3. what is the main purpose of the conducted research, one gets the impression that it is a compilation of a dozen or so articles. It should be emphasized that it is well done.
4. Authors should systematize their work and put more emphasis on their research.
Author Response
Comments and suggestions
The article is an extensive material on polymer composites. The authors analyze the extensive literature. concerning both the theory and practice related to the production of composites and their properties. The literature contains 113 items. Despite the great value of the work, I would like to note that a similar study has already been done, there is also a book study. My doubts are whether the journal was correctly selected for the publication of this type of article. Here my comments:
- What is the thesis of the work, what is the specific goal.
Reply:
mode I fatigue delamination is a dominant failure mode in composites. The lower values of interlaminar fatigue loads lead to complete collapse of composite structures, and consequently it can be life-threatening structural reliability problem. In carbon fiber reinforced epoxy composites this issue becomes more thought-provoking since the cyclic cracks’ propagation cannot be sensed using visual inspection. With increasing demand of composites in critical structural components, it is essential to ensure damage tolerance to increase safety. Considering this we have given a comprehensive review where we have described the fatigue behavior of various reinforcements as well as the different factors affecting the mode I fatigue performance of composites are given. Approaches for composites life enhancement against mode I fatigue loading have also been summarized, which could develop a well-rounded understanding of mode I fatigue behaviors of composites and thus, it can help engineers to design composites with higher interlaminar strength.
- The topic is not entirely original, but it presents a great substantive value related to the development of composites. The main goal should be edited and the research carried out more emphasized, so that the reader can clearly know what has been achieved in the work.
Reply:
In this review articles we have tried to review all relevant articles on mode I fatigue delamination od composites that could help in development of composite structures with improved fatigue performance. We have edited the ain goal of this article as per reviewer comments.
- what is the main purpose of the conducted research, one gets the impression that it is a compilation of a dozen or so articles. It should be emphasized that it is well done.
Reply: as per reviewer comment we have revised the main goal of this article for better understanding of readers.
- Authors should systematize their work and put more emphasis on their research.
Reply: Done
Reviewer 3 Report
1. The paper is a comprehensive review on fiber reinforced polymeric composites and their fatigue behavior.
2. The topic is original and relevant in the field? It presents a detailed review in the field?
3. Compared with other published material, it provides a somehow overall idea of the state-of-art in the field of fiber reinforced composites?
4. There is no methodology. as it is a review. Further, the most recent literature should be included.
5. The summarized conclusions are consistent with the evidence and arguments presented and they address the main question posed?
6. There are a lot of references but the most recent literature in the last 5 years should be emphasized.
7. Tables and figures must be presented in best possible quality, resolution and with reference to the original source.
8. English editing needed.
9. Some recent literature should be added.
10. Figures should be presented in better quality.
11. Further details of fiber types and effects on bio-composites can be extended to natural fibers.
Author Response
Comments and suggestions
- The paper is a comprehensive review on fiber reinforced polymeric composites and their fatigue behavior.
The topic is original and relevant in the field? It presents a detailed review in the field?
Reply: yes we have given details in our articel
- Compared with other published material, it provides a somehow overall idea of the state-of-art in the field of fiber reinforced composites?
Reply: Yes, we have tried best to our knowledge.
There is no methodology. as it is a review. Further, the most recent literature should be included.
Reply: we have the most recent literature in the article.
The summarized conclusions are consistent with the evidence and arguments presented and they address the main question posed?
Reply: Yes, thank you
- There are a lot of references but the most recent literature in the last 5 years should be emphasized.
Reply: We have focused on most recent literature.
- Tables and figures must be presented in best possible quality, resolution and with reference to the original source.
Reply: We have redrawn some of the figures but some graphs and SEM images are given as they were in the original source. We have tried our best to give possible quality and high resolution of images.
- English editing needed.
Reply: Done
- Some recent literature should be added.
Reply: Recent literature has been added in the revised article.
- Figures should be presented in better quality.
Reply: We have redrawn some of the figures, but some graphs and SEM images are given as they were in the original source. We have tried our best to give possible quality and high resolution of images.
- Further details of fiber types and effects on bio-composites can be extended to natural fibers.
Reply: This review article scope is structural composites so we considered the fatigue behavior of high performance fiber reinforced composites such as glass and carbon based composites, these have limited use as bio composites, so we did not focused on natural fiber reinforced composite, in future a separate detailed article can be written to review the fatigue behavior of natural fiber based bi-composite
Round 2
Reviewer 2 Report
The introduced corrections supplement the knowledge and explain the more posed problem.